# Plant-Wide Target Metabolomics Provides a Novel Interpretation of the Changes in Chemical Components during *Dendrobium officinale* Traditional Processing

**DOI:** 10.3390/antiox12111995

**Published:** 2023-11-12

**Authors:** Pengfei Liu, Bei Fan, Yuwen Mu, Litao Tong, Cong Lu, Long Li, Jiameng Liu, Jing Sun, Fengzhong Wang

**Affiliations:** Institute of Food Science and Technology, Chinese Academy of Agricultural Sciences, Key Laboratory of Agro-Products Processing, Ministry of Agriculture and Rural Affairs, Beijing 100193, China; liupengfei0424@zidd.ac.cn (P.L.); fanbei@caas.cn (B.F.); muyw@gsagr.cn (Y.M.); tonglitao@caas.cn (L.T.); lucong@caas.cn (C.L.); lilong02@caas.cn (L.L.); liujiameng@caas.cn (J.L.)

**Keywords:** *Dendrobium officinale*, traditional processing, widely targeted metabolic analysis, chemical compositions, antioxidant activity

## Abstract

The traditional processing of *Dendrobium officinale* (DO) is performed in five necessary processing steps: processing fresh strips, drying at 85 °C, curling, molding, and drying at 35 °C (Fengdou). The antioxidant activity of DO is increased after it is processed into Fengdou. To comprehensively analyze the changes in the functional components, a plant-wide target metabolomics approach was employed. In total, 739 differential chemical components were identified in five processing treatments, mainly highlighting differences in the levels of phenolic acids, flavonoids, lipids, and amino acids and their derivatives, and the glycosylation of aglycone resulted in the upregulation of flavonoid glycoside levels. Temperature is a key factor in DO processing during production. In addition, the enrichment of specific differential chemical components was found mainly in five different metabolic pathways: glucosinolate biosynthesis, linoleic acid metabolism, flavonoid biosynthesis, phenylpropanoid biosynthesis, and ubiquinone and other terpene quinone biosynthesis. A correlation analysis clarified that total phenols and flavonoids show a significant positive correlation with antioxidant capacity. This study provides new insights into the influence of the processing processes on DO quality, which may provide guidance for the high-quality production of DO.

## 1. Introduction

*Dendrobium officinale* (DO) Kimura and Migo is a perennial herb in the *Dendrobium* genus of the family Orchidaceae, which has been used historically for edible and medicinal purposes for over 2000 years [1]. Previous studies have shown that DO has excellent medicinal qualities, benefiting the stomach, generating fluid, nourishing “Yin” and clearing heat, treating gastric ulcers, and treating chronic atrophic gastritis associated with Yin deficiency [2]. The types of chemical components contained in the plant mainly include polysaccharides, phenols, alkaloids, and amino acids [3,4], and there is substantial in vitro or in vivo evidence showing that the bioactive compounds present in DO have antioxidant properties [5] that have attracted scholars to conduct in-depth research on DO. Among them, polysaccharides have been reported to reduce insulin resistance and abnormal lipid metabolism in obese mice [6] and inhibit inflammatory factors from alleviating liver metabolic disorders in diabetic mice [7]. While some studies have established the immunomodulatory effects of DO alkaloids [8], there is growing evidence that the phenolic compounds in DO have antioxidant and cytoprotective properties, and bisbenzyl compounds have been widely used in the production of various skin care products and drugs [9]. DO can provide novel bioactive compounds and natural ingredients for the agri-food, cosmetic, and pharmaceutical industries to replace synthetic compounds, as well as value-added plant by-products.

The growing sophistication of farming techniques will result in the oversupply of DO, and prices will decline. Processing and storage methods are also receiving attention. The moisture content of fresh DO can be up to 80% [10], and though it appears shiny, it is easily infected with breeding bacteria. Thus, it is necessary to control moisture during DO post-harvest storage, transportation, and processing without a biological activity reduction. The common drying methods for DO include sun drying, hot air drying, infrared drying, freeze drying, etc. However, Fengdou has always been a popular processing product of DO due to its advantages of small space and good storage qualities. Our previous studies found that the total flavonoid and total polyphenol contents of DO were elevated after it was processed into Fengdou [11], but the exact mechanism remains unclear. Several studies have been conducted to compare the chemical components before and after DO processing, such as polysaccharides, total phenols, and total flavonoids [12]. However, Fengdou is made through various processes, such as drying, hooping, and baking, and the mechanisms of compositional changes during this process are largely unknown. Hence, it is critical to systematically compare the chemical profiles during the processing of fresh DO strips into Fengdou.

Plant-wide target metabolomics is a new field of metabolomics, which can be effective in analyzing high-dimensional complex data and help to observe the differences between samples. Significant variations in the metabolic profile of DO have been revealed by metabolomic approaches, which depend on a variety of factors, such as the species, growth conditions, and plant parts [13]. There are also studies that reveal the material basis and mechanisms of disease prevention and treatment through metabolomics, such as the application of DO to gastric mucosal protection [14] and the study of metabolic mechanisms of its preventive effects on diabetes [15]. Although research has provided lots of information about DO products under different conditions, the compositional changes in DO during processing have not been clarified.

The main objective of this work is to comprehensively study the dynamic changes in compounds during processing through metabolomics and to summarize the key processing steps. This will help in further investigating the accumulation and metabolic regulation mechanisms of active compounds in DO during processing and provide new insights for quality control during the production of functional food from DO.

## 2. Materials and Methods

### 2.1. DO samples in Different Processing Stages

In this study, 15 kg of fresh DO was collected on 10 November 2021 from the Longmu *Dendrobium* Department of Longshan County, Longshan Town (Yunnan, China), and authenticated by Prof. Pengfei Tu according to the features described in the *Chinese Pharmacopoeia* (2020 edition). Processing was carried out according to local DO standards for food safety. Briefly, the 15 kg of fresh DO was cleared of debris (residual surface leaves and other debris were removed, and irregular branches were excised), washed with pure water, and dried to control the moisture, and then 1 kg was randomly sectioned off as sample FA1. The remaining DO was placed in an oven at 85 °C for baking and was rubbed while drying to facilitate curling. Residual leaf sheaths and other debris were removed as much as possible during the softening process, and water was removed until the DO softened. At this stage, 1 kg was taken out as sample FB2. The remaining part of the baked soft *Dendrobium* strips was twisted into a spiral shape, and 1 kg of the sample was left, named sample FC3. After shaping twice, the sample was tightly curled, not spread out, with a beautiful, uniform shape, and 1 kg of the sample was named sample FD4. Finally, it was placed in the oven at 35 °C, dried at low temperature, and named sample FE5, as shown in Figure 1. The five samples were used for UPLC-ESI-MS/MS analysis, and the analysis was performed in triplicate for each sample.

### 2.2. Sample Preparation and Extraction

The five types of samples were freeze-dried, and then ground with a mixer mill (MM400, Germany Retsch, Haan, Germany) and zirconia beads for 1.5 min at 30 Hz. The lyophilized powder (50 mg) was dissolved with 1.2 mL 70% methanol solution, and vortexed every 30 min for 30 s (6 times in total). The samples were then centrifuged for 10 min at 12,000 rpm, and the supernatant was taken for activity evaluation. Meanwhile, other supernatants were filtrated prior to UPLC-ESI-MS/MS analysis. Moreover, 100 μL aliquots of each sample were pooled to yield a quality control (QC) sample. The QC sample was performed every six samples to assess the repeatability of the measurement process.

### 2.3. Antioxidant Activity

#### 2.3.1. DPPH Radical Scavenging Activity Assay

The experiments were performed as modified on the protocol described by Gong et al. [16]. Firstly, a proper amount of the extraction filtrate was obtained by the method in Section 2.2 to prepare samples with different concentrations. Then, 2 mL of DO sample solutions at different concentrations were added into 2 mL DPPH free radical stock solution (0.1 mg/mL) separately, after which the mixed solution were shaken well and left in the dark for 30 min. At the same time, anhydrous 95% methanol was used instead of the sample extract as the negative control, and Trolox was used as a positive control. The absorbance was measured at 515 nm. Each sample was measured three times. The percentage of inhibition was calculated
(1)$$\left[AC-\frac{AS}{AC}\times 100\right]$$
where *AC* is the absorbance of the negative control and *AS* is the absorbance of the test samples.

#### 2.3.2. ABTS Radical Scavenging Activity Assay

The antioxidant capacity was evaluated via the 2,2′-azino-bis (3-ethylbenzthiazoline-6-sulfonic acid) (ABTS) radical method reported by Liu et al. [17]. The calculation method employed was the same as that described in Section 2.3.1.

### 2.4. Total Phenolics and Total Flavonoid Quantification

The total phenolic content (TPC) was determined using the Folin–Ciocalteu method described by Castaldo [18], and the results are expressed as mg of gallic acid equivalent (GAE) per or 100 g of dry matter (DO). In addition, the total flavonoid content (TFC) was determined using the colorimetric method described by Luo et al. [19], and the results are expressed as mg of rutin equivalent (CE) per or 100 g of DO.

### 2.5. UHPLC-MS Conditions

The analytical conditions were as follows, UPLC: column, Agilent SB-C_18_ (1.8 µm, 2.1 mm × 100 mm). The mobile phase was composed of pure water with 0.1% formic acid (A) and acetonitrile with 0.1% formic acid (B). The sample measurements were performed with a gradient program that employed the starting conditions of 95% A and 5% B. Within 9 min, a linear gradient to 5% A, 95% B was programmed, and a composition of 5% A, 95% B was kept for 1 min. Subsequently, a composition of 95% A, 5.0% B was adjusted within 1.1 min and kept for 2.9 min. The flow velocity was set as 0.35 mL per minute, the column oven was set as 40 °C, and 4 μL aliquots were injected into the column. The effluent was alternatively connected to an ESI-triple quadrupole-linear ion trap (Q TRAP)-MS. An AB4500 Q TRAP UPLC/MS/MS System, equipped with an ESI Turbo Ion-Spray interface, operating in positive and negative ion mode and controlled using Analyst 1.6.3 software (AB Sciex, Framingham, MA, USA) was used. The ESI source operation parameters were as follows: ion source, turbo spray; source temperature 550 °C; ion spray voltage (IS) 5500 V (positive ion mode)/−4500 V (negative ion mode); ion source gas I (GSI), gas II(GSII), curtain gas (CUR) was set at 50, 60, and 25 psi, respectively; the collision-activated dissociation (CAD) was high.

### 2.6. Qualitative of Chemical Compositions

The compounds were identified based on the MWDB (metwa database). The characterization of substances was performed based on secondary spectral information, and the analysis removed isotopic signals and repeating signals containing K^+^, Na^+^, NH_4_^+^, and repeating signals of fragment ions that are themselves other larger molecular weight substances. The quantitative analysis was mainly based on their abundance in the UPLC-ESI-MS/MS analysis. The precursor ions were fragmented by collision chamber-induced ionization, and the resulting fragment ions were filtered through the triple-quadrupole rod to select the desired characteristic fragment ions, eliminating non-target ion interference and enhancing the accuracy and reproducibility of quantification.

### 2.7. Statistical Analysis of Chemical Compositions

To study the accumulation of differences in metabolite, a multivariate statistical analysis was conducted on the metabolic data of each sample. Principal component analysis (PCA) was employed to visualize the variables, and was performed using GraphPad Prism v9.01 (GraphPad Software Inc., La Jolla, CA, USA). An OPLS-DA model was performed using the R software package MetaboAnalystR 3.0 to compare the composition characteristics of different DOs. The variable importance in the projection (VIP) ≥ 1 in the OPLS-DA model and the absolute FC (fold change) ≥ 2 or FC ≤ 0.5 was set for screening differential chemical composition. Venn diagrams were generated online to show the number of differential chemical compositions. The KEGG compound database was used to annotate the different chemical compositions, which were mapped to the KEGG pathway database. The pathways with compositions that were significantly regulated were subjected to component pathway sets enrichment analysis. *p*-values from hypergeometric tests are used to assess their significance.

## 3. Results and Discussion

### 3.1. Effect of DO Samples on Antioxidant Activity

To study the effect of processing on the antioxidant capacity of DO, we measured DPPH free radical scavenging activity and ABTS cation free radical scavenging activity (Figure 2A). The samples of FE5 showed the strongest antioxidant activity in both DPPH and ABTS assays. The scavenging effect of DPPH radicals was significantly enhanced with the increase in DO extract concentration in the range of 3.125–25.000 mg/mL. In the same bolus, the DPPH radical scavenging effect also gradually increased during the processing of DO. At the concentration of 3.125 mg/mL, the DPPH radical scavenging rates for samples FA1 and FE5 were 25.33% and 32.99%, respectively, indicating a superior oxidation resistance of FE5 compared to FA1. The same results were also confirmed in the ABTS radical scavenging experiment (Figure 2B). The increase in antioxidant capacity in the processing steps could be due to the formation of Maillard reaction products (MRPs) [20] under high temperature and the promotion of intracellular secondary metabolite release under the external pressure.

### 3.2. Overview of the Profile of Composition

To explore the material basis of antioxidant activity of DO, the TPC and TFC levels were tested in each sample. As shown in Figure 2C,D, the content of both TPC and TFC increased significantly from FA1 to FE5. The TPC and TFC observed in the FE5 were quantified at values of 4.25 mg GAE/g and 0.44 mg CE/g, respectively. It is demonstrated that antioxidant effects are closely related to polyphenol content, such as in blueberries, fragaria nubicola, apples, and other vegetables or fruits [21,22,23]. Therefore, a widely non-targeted metabolite analysis based on an UPLC–MS/MS system was employed to discover the metabolic profiling of FA1~FE5. The composition characteristics of each sample were screened and a total of 1182 chemical compositions (Appendix A) were determined, including alkaloids (8.12%), amino acids and derivatives (10.41%), flavonoids (16.50%), lignans and coumarins (3.81%), lipids (13.45%), nucleotides and derivatives (5.24%), organic acids (8.29%), phenolic acids (13.37%), quinones (3.47%), terpenoids (4.23%), and others (13.11%) (Figure 3A). The results show that flavonoids, phenolic acids, lipids, amino acids, and derivatives were the dominant chemical compositions in DO.

### 3.3. Multivariate Statistical Analysis

A multivariate statistical analysis was performed on the five groups to show differences in their chemical composition. A principal component analysis was performed using unsupervised models to analyze the differential chemical components and assess the overall differences between each sample (Figure 3B). In the score plot, the two principal components explained 58.4% of the total variance, and the first principal component (PC1) and the second principal component (PC2) indicated sufficient separation between samples of different groups. At the same time, PC1 and PC2 accounted for 37.86% and 20.54% of the variance, respectively. These results indicate that the processing of DO samples strongly influences the metabolite profiles in different ways. It can also be seen from the figure that samples FA1 and FB2 were clearly separated, as well as FD4 and FE5. However, there was no significant difference between FB2, FC3, and FD4. It is tentatively suggested that the processing steps from FA1 to FB2 and FD4 to FE5 may be the key stages affecting the compositional changes. The OPLS-DA analysis model was another favorable means used to screen for differential chemical compositions by removing effects that are not relevant to the study. In this experiment, the OPLS-DA score plots of each group (Appendix A) show that the five samples of different processing processes were separated in pairs, indicating the variability of metabolic profiles among sample groups. Based on the results of the OPLS-DA model, 200-times permutation tests were performed to validate the accuracy of the OPLS-DA model in this experiment (Appendix A). As shown in Figure 3C, the correlation coefficients between the different samples during the processing are all greater than 0.7, especially for FA1 and FB2, which have the least correlation among the adjacent groups, indicated that baking at 85 °C has a greater effect on the changes in chemical composition. Similarly, the effect of curling and sizing on the chemical composition of the samples was inappreciable. It can be seen that DO has significant variability in chemical compositions between the raw material and the processed product Fengdou, with the highest contribution from baking at 85 °C, followed by drying at 35 °C, and smaller effects from curling and shaping.

### 3.4. Identification of Differential Chemical Composition

According to the results of the OPLS-DA model, a total of 739 differential chemical compounds were screened between groups for variable importance (VIP) ≥ 1, with a fold change (FC) ≥ 2 for upregulated compounds and FC ≤ 0.5 for downregulated compounds in the prediction. The differential chemical compositions were mainly divided into amino acids and derivatives, phenolic acids, flavonoids, lipids and others, with a large proportion of lipids (17.19%), amino acids and derivatives (14.21%), phenolic acids (13.66%) and flavonoids (10.82%). According to the volcano plots, the fewest differential metabolites were observed in the FC3 and FD4 group (Figure 4C), with 67 (30 up and 37 down). Moreover, there were 481 significantly different chemical compositions between FA1 and FB2 (271 up and 210 down) (Figure 4A), 195 ones between FB2 and FC3 (166 up and 29 down) (Figure 4B), and 362 ones between FD4 and FE5 (51 up and 311 down) (Figure 4D). From Figure 4E, it can be clearly seen that there are 80 unique differential chemical compounds in the FA1 vs. FB2 group. Compared with other groups, the FA1 vs. FB2 group had more unique and differentiated chemical components, further confirming the hypothesis that temperature is a key factor affecting changes in chemical composition.

#### 3.4.1. Changes in Phenolic Acids Profile

Phenolic acids are one of the main components of DO and are commonly found in natural plants. As shown in the correlation heat map, there is a significant correlation between total polyphenols and antioxidant capacity (Figure 5). In total, 68 differential phenolic acid compounds were screened in the DO FA1 vs. FE5 group (Appendix A), with 31 up-regulated and 37 down-regulated. During the processing of DO, there were 21 unique differentially expressed compounds in the FA1 vs. FB2 group (Table 1). In the FA1 vs. FB2 group, 6 differential chemical compositions such as 3,4,5-trimethoxybenzoic acid were down-regulated, while 15 differential chemical compositions such as 4-hydroxybenzaldehyde were up-regulated. Among them, vanillin (Figure 6A) and 4-hydroxybenzaldehyde were up-regulated after roasting processing at 85 °C, which results in the existence of aromas increasing, and these changes may be closely related to lignans. Studies have indicated that lignans are widely distributed in plants, which can be oxidized to form vanillin under high-temperature conditions [24]. In the correlation analysis, vanillin had strong antioxidant capacity that reacted with ABTS radicals via a self-dimerization mechanism [25]. In addition, the phenolic acid chemical compositions such as syringaldehyde-4-*O*-glucosideand 2-*O*-galloyl-D-glucose that contain glycosides were also upregulated after baking at 85 °C. The fold changes (Log_2_FC) of some glycosides vary greatly, for example, 2-*O*-galloyl-D-glucose and syringaldehyde-4-*O*-glucoside increased in orders of 15.80 and 13.04, respectively (Figure 6B). Meanwhile, the content of syringaldehyde showed a downward trend from the FA1 to FB2 processes. Yu et al. [26] reported the presence of glycosylation modifying enzymes and pathways of phenolic acids in poplar, and accordingly, the above phenolic glycosides may be formed from phenolic acids modified by glycosides. In the study of glycosides synthesis by Chen et al. [27], temperature was an important factor in enhancing the conversion rate of chemical compositions. The derivatives’ differential metabolites in the FD4 vs. FE5 group included seven species of vanillic acid, methyl syringate (Figure 6D), and 4-hydroxybenzoic acid, which showed a downward trend. Meanwhile, the relative content of syringic acid and *p*-coumaric acid were increased in FD4 compared with FE5 (Figure 6E). Drying at 35 °C greatly influences the content of compounds with carboxyl and ester bonds, which may be the reversible reaction catalyzed by esterase and the decrease of ester content due to high acidity [28]. No specific differential phenolic acid compounds were produced during the curling and shaping processes. In addition, coniferyl alcohol was a persistent differential metabolite that was down-regulated in the curling process (Figure 6F). Coniferyl alcohol, consistently observed as a downregulated differential metabolite during the curling process, was upregulated after the fixation step, aligning with findings from Ti et al. [29] regarding extruded black rice.

#### 3.4.2. Changes in Flavonoid Profile

The biosynthetic pathway of flavonoids in plants is complex, with most of them existing in the form of flavonoid-bound glycosides, while a few exist in free form. A total of 35 different flavonoid chemical compounds were screened in the FA1 vs. FE5 group, resulting in the up-regulation of 31 compounds and the down-regulation of 4 (Appendix A). Among these, 17 flavonoids that include naringenin, quercetin-3,7-di-*O*-glucoside, naringenin-4′-*O*-glucoside, and phloretin-4′-*O*-glucoside (Figure 6G) were up-regulated as unique differential chemical compounds in the FA1 vs. FB2 group (Table 1).

Due to the loss of water during baking at 85 °C, glycoside hydrolysis reactions are almost non-existent. It is widely known that enzyme and moisture are two key factors in glycoside hydrolysis, such that the first barking has the effect of deactivating enzymes and protecting glycosides [30]. The naringenin, quercetin-3,7-di-*O*-glucoside, naringenin-4′-*O*-glucoside, phloretin-4′-*O*-glucoside, and other flavonoid glycoside chemical compounds are positively correlated with the radical scavenging ability of ABTS in the process from FA1 to FB2. It is documented that TFC prevents the accumulation of lipofuscin and protein carbonylation, and upregulated the gene expression levels of hsp-16.2, gst-4, etc. [31]. In this experiment, no specific differential flavonoid compounds were found during the curling and shaping processes in the processing of DO. However, 4′-*O*-glucosylvitexin and isosaponarin were common differential chemical compositions (Figure 6H,I), with some being down-regulated during the curling process and up-regulated during the shaping step. This phenomenon is consistent with the findings from twin-screw extrusion of Folium Artemisiae Argyi and puffed cereals [32,33].

#### 3.4.3. Changes in Amino Acids and Derivatives Profile

In the current study, a total of 88 amino acids were identified in the FA1 vs. FE5 group (Appendix A), with 69 metabolites down-regulated and only 19 up-regulated, showing an overall trend of down-regulation. This indicates an overall down-regulation trend. Amino acids can be categorized into essential and non-essential types. Among the essential amino acids, L-threonine, L-isoleucine, L-leucine, L-valine, L-phenylalanine, L-methionine, L-lysine, and L-tryptophan exhibited down-regulation during the processing of DO, influenced by baking at 85 °C and drying at 35 °C. Furanosine is a product of the L-lysine Maillard reaction [34]. When the internal temperature of grilling and frying is lower than 90 °C, lysine is mainly synthesized as furanosine [35]. The important condition for significant changes in 13 non-essential amino acids such as γ-glutamyl phenylalanine, L-aspartic acid and *N*-methylglycine during DO processing was 85 °C drying. Meanwhile, 16 non-essential amino acids were significantly down-regulated during drying at 35 °C. During heat processing, the total concentration of free amino acids in the heated samples was significantly lower than that in the raw sample, consistent with the amino acid changes observed after temperature treatment in this experiment [36].

#### 3.4.4. Changes in Lipids Profile

Based on the dynamic changes in lipid during the processing of DO, 118 lipid differential chemical compounds were screened from the FA1 vs. FE5 group. Among them, only half were up-regulated (Appendix A). Lipids can be further categorized into lysophosphatidylcholine (LPC), lysophosphatidylethanolamine (LPE), glycerol ester (GE), free fatty acids (FC), phosphatidylcholine (PC), and 3-Hydroxy-palmitic acid methyl ester. There were 15 FCs such as hydroperoxylinoleic acid and 3-hydroxy-palmitic acid methyl ester, 4 LPEs such as LPE 17:0 and LPE 16:0, and 11 LPCs such as LPC 20:2 and LPC 12:0 in DO processing as the specific differential chemical compositions in the FA1 vs. FB2 group. In the FA1 vs. FB2 group, FCs were down-regulated, while LPCs and LPEs were up-regulated. Additionally, 13 down-regulated FCs, including γ-linolenic acid and α-hydroxy linoleic acid, were also present in the FD4 vs. FE5 group as specific differential chemical compounds. The decrease in FCs may be partly due to the inactivation of lipase (LA) and lipoxygenase (LOX) caused by high-temperature processing [37], which catalyzed the hydrolysis of triacylglycerols (TAGs) to release FC [38]. It is worth noting that the inactivation of LA and LOX is influenced by two key factors, namely temperature and time [39], which is consistent with DO 85 °C baking and 35 °C drying in this study. This indicates that heating is detrimental to the retention of FC. Although curling and setting did not cause specific differential chemical compositions, some GEs were affected by them and showed an upregulation trend. Yang et al. [40] suggested that the bound lipid content of noodles continues to increase during the pressing process, with the most important GE of pressed oil exceeding leached oil by 30%. This indicates that extrusion facilitates lipid retention.

### 3.5. KEGG Function Annotation and Enrichment Analysis

The differential chemical compositions were analyzed via KEGG metabolic pathways separately. Firstly, the 88 amino acids and their derivatives were annotated into 55 metabolic pathways. Among these, the six essential amino acids were significantly enriched in glucosinolate biosynthesis (ko00966) (Appendix A). Meanwhile, 118 differential lipids chemical compositions were annotated into 10 metabolic pathways (Appendix A), with alpha-Linolenic acid metabolism (ko00592) and Linoleic acid metabolism (ko00591) being the two most significantly enriched pathways (Appendix A). These pathways are associated with the synthesis of oxylipins in fatty acid metabolism, which can promote lipid metabolism. The phenolic acids and flavonoids as secondary chemical compositions, and 68 differential phenolic acids were annotated to 13 metabolic pathways (Appendix A), with 3 metabolic pathways more significantly enriched in phenylalanine metabolism (ko00360), phenylpropanoid biosynthesis (ko00940), and ubiquinone and other terpenoid quinone biosynthesis (ko00130). This is the same result as the enrichment pathway for some of the flavonoids studied by Xue et al. [41].

### 3.6. Processing-Induced Chemical Transformation Mechanisms

Generally, the processing-induced changes in the composition of DO are primarily driven by factors such as temperature, water, and external forces. These factors also reveal the chemical transformation mechanisms that occur during processing [42,43]. With regard to the shift of differential chemical composition, many reactions may be involved, such as oxidation, hydrolyze, esterification of small-molecular compounds as well as condensation, glycosylation, etc. According to the metabolomics results, the differential chemical compositions of flavonoids were taken as an example to briefly describe the changes (Figure 7). Firstly, the reaction for glycoside hydrolysis was disrupted by a sharp decrease in moisture content, and the hydrolase enzymes gradually lost their activity, which contributes to retaining components. The contents of aglycones such as quercetin and phloretin showed a downward trend in the FA1 vs. FB2 group. At the same time, glycosides (quercetin-3,7-di-*O*-glucoside and phloretin-4′-*O*-glucoside) were up-regulated in FA1 vs. FB2, suggesting that aglycones were converted into glycosides by polysaccharide or monosaccharide modification. This also indicates that the rate of glycosylation was greater than de-glycosylation when baking at 85 °C. This suggests that a broad range of flavonoid substrates could be glycosylated at their 3- and/or 7-hydrogen sites by the recombinant enzyme, such as quercetin [44]. Furthermore, some phenolic hydroxyl and ester, as well as glycosides containing glycosidic bonds, might involve oxidative decomposition, and esterification of alcoholic hydroxyl during processing. These processes may represent potential chemical mechanisms explaining the differences between raw materials and processed DO.

## 4. Conclusions

In this study, the highest antioxidant activity was observed in FE5 during the various stages of DO traditional processing. Liquid-mass-spectrometry-based metabolomics techniques were employed to identify changes in chemical composition during the traditional processing of fresh strips to DO. The main factors leading to the changes in compositions during the processing were baking at 85 °C and drying at 35 °C, with less effects from curling and setting. Phenolic acids, flavonoids, lipids and amino acids and their derivatives changed most significantly during the baking at 85 °C and drying at 35 °C. Phenolic acids, flavonoids, and lipids produced 28, 20 and 45 unique differential chemical compounds, respectively. Additionally, the eight essential amino acids, which were affected by the baking at 85 °C and drying at 35 °C, and the changes in their contents may affect the quality and activity of DO. *P*-coumaric acid, a unique differential metabolite and a natural antioxidant, connected glucosinolate biosynthesis, flavonoid biosynthesis, phenylpropanoid biosynthesis, ubiquinone and other terpenoid-quinone biosynthesis, and other terpenoid-quinone metabolic pathways. By precisely regulating the key chemical compositions in metabolic pathways, the material basis for the antioxidant capacity of DO was established. This study provides a valuable reference for the quality control and enhancement of active chemical components during the processing of DO.

## Figures and Tables

**Figure 1 antioxidants-12-01995-f001:**

Appearances of DO samples at the end of each processing step. The fresh strip (FA1), drying at 85 °C (FB2), curling (FC3), molding (FD4), and drying at 35 °C (FE5).

**Figure 2 antioxidants-12-01995-f002:**
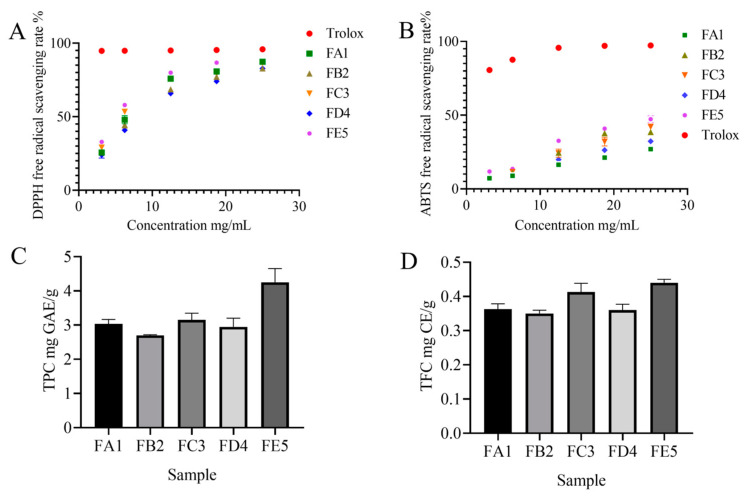
The DPPH scavenging capacity (**A**) and ABTS scavenging capacity (**B**) of DO samples in different processes; the contents of total phenols (**C**) and flavonoids (**D**) of DO samples in different processes.

**Figure 3 antioxidants-12-01995-f003:**
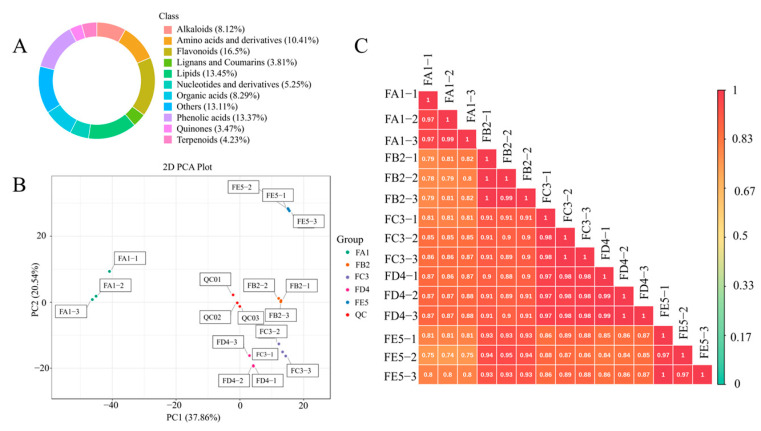
Classification of the 1182 composition of DO samples (**A**); PCA score plot (**B**). The sampling groups were color-coded as follows: green, FA1; orange, FB2; purple, FC3; pink, FD4; blue, FE5; red, QC. Correlation diagram of composition of different processes of DO (**C**).

**Figure 4 antioxidants-12-01995-f004:**
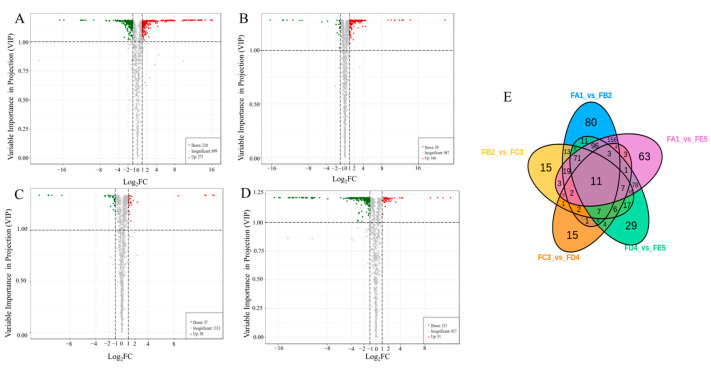
Volcano plots of the differential ingredient expression levels between FA1 vs. FB2 groups (**A**), the FB2 vs. FC3 groups (**B**), the FC3 vs. FD4 group (**C**) and the FD4 vs. FE5 group (**D**), red indicating upward and green indicating downward; Venn diagram illustrating the overlapping and specific differential compositions for four comparison groups (FA1 vs. FB2, FB2 vs. FC3, FC3 vs. FD4, FD4 vs. FE5) (**E**).

**Figure 5 antioxidants-12-01995-f005:**
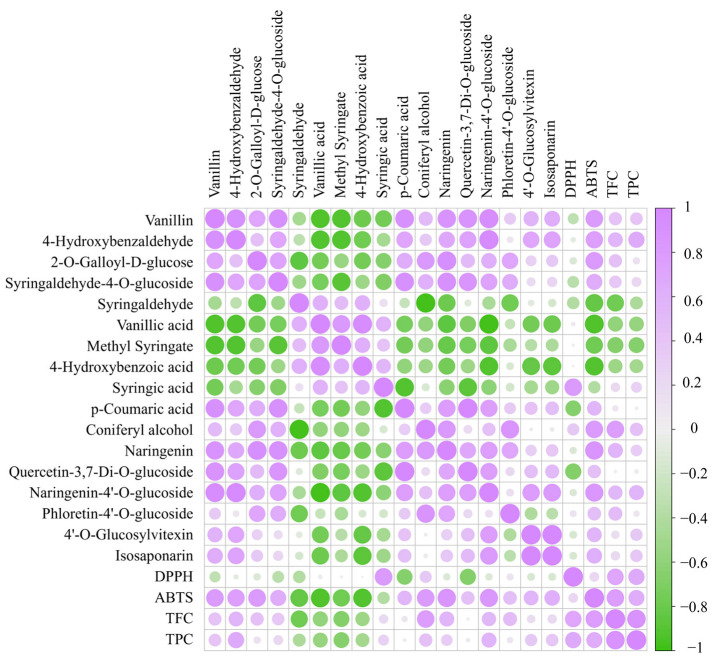
Correlation heatmap of specific differential compositions and antioxidant activity. TPC: total phenolic content; TFC: total flavonoid content; DPPH: DPPH·scavenging capacity; ABTS: ABTS + scavenging capacity.

**Figure 6 antioxidants-12-01995-f006:**
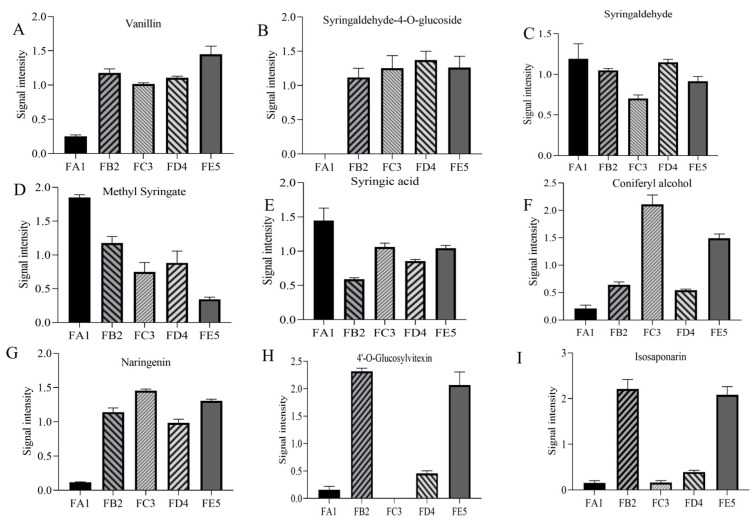
Differences in the relative contents of the main differential chemical composition in the different processing procedures of DO. To reduce the influence of concentration on pattern recognition, the peak area matrix is normalized. Bar charts show the relative content of vanillin (**A**), syringaldehyde-4-*O*-glucoside (**B**), syringaldehyde (**C**), methyl syringate (**D**), syringic acid (**E**), coniferyl alcohol (**F**), phloretin-4′-*O*-glucoside (**G**), 4′-*O*-glucosylvitexin (**H**), and isosaponarin (**I**).

**Figure 7 antioxidants-12-01995-f007:**
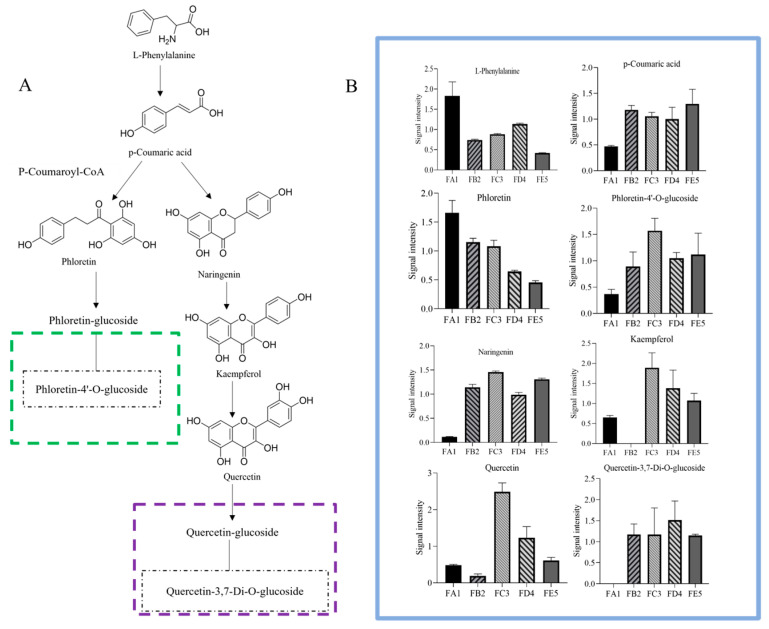
The alteration of the flavonoid pathway (**A**) and relative amount (**B**) during the processing of DO. To reduce the influence of concentration on pattern recognition, the peak area matrix is normalized in relative amount (**B**).

**Table 1 antioxidants-12-01995-t001:** Unique differential chemical compositions (VIP ≥ 1) responsible for the chemical ingredient variation caused by the traditional processing of DO.

No.	Compounds	Formula	*m*/*z*	Type	Fold Change	VIP	*p*-Value
FA1 vs. FB2							
1	2-*O*-Galloyl-D-glucose	C_13_H_16_O_10_	332.2601	PD	6.02 × 10^4^ ↑	1.185	0.001 × 10^−1^
2	Gallacetophenone	C_8_H_8_O_4_	168.1467	PD	1.23 × 10^4^ ↑	1.185	0.001 × 10^−1^
3	Dihydrocaffeoylglucose	C_15_H_20_O_9_	344.3139	PD	9.27 × 10^3^ ↑	1.185	0.002
4	Syringaldehyde-4-*O*-glucoside	C_15_H_20_O_9_	344.3139	PD	8.44 ↑	1.185	0.005
5	Vanillin	C_8_H_8_O_3_	152.1473	PD	4.52 ↑	1.181	0.003 × 10^−1^
6	Methyl 4-hydroxybenzoate	C_8_H_8_O_3_	152.1743	PD	4.82 ↑	1.181	0.001 × 10^−2^
7	1-*O*-Feruloyl-*β*-D-glucose	C_16_H_20_O_9_	356.3246	PD	12.2 ↑	1.180	0.001 × 10^−1^
8	*p*-Coumaric acid	C_9_H_8_O_3_	164.158	PD	2.51 ↑	1.178	0.003
9	Isovanillin	C_8_H_8_O_3_	152.1473	PD	4.52 ↑	1.174	0.009
10	1-*O*-Sinapoyl-*β*-D-glucose	C_17_H_22_O_10_	386.3506	PD	2.89 ↑	1.173	0.007
11	Sinapinaldehyde	C_11_H_12_O_4_	208.2106	PD	2.47 ↑	1.172	0.001 × 10^−1^
12	2-(Formylamino)benzoic acid	C_8_H_7_NO_3_	165.1461	PD	2.48 ↑	1.167	0.002 × 10^−1^
13	4-Hydroxybenzaldehyde	C_7_H_6_O_2_	122.1213	PD	2.14 ↑	1.156	0.003
14	1-*O*-(3,4-Dihydroxy-5-methoxy-benzoyl)-glucoside	C_14_H_18_O_10_	346.2867	PD	0.27 ↓	1.151	0.036
15	Caffeic aldehyde	C_9_H_8_O_3_	164.158	PD	2.04 ↑	1.106	0.004
16	Cinnamic acid	C_9_H_8_O_2_	148.1586	PD	6.44 ↑	1.168	0.002
17	2-Acetyl-3-hydroxyphenyl-1-*O*-glucoside	C_15_H_20_O_7_	312.3151	PD	0.35 ↓	1.166	0.013
18	Vanillin acetate	C_10_H_10_O_4_	194.184	PD	0.33 ↓	1.135	0.05
19	4-*O*-Methylgallic Acid	C_8_H_8_O_5_	184.1461	PD	0.41 ↓	1.102	0.003
20	Eudesmic acid	C_10_H_12_O_5_	212.1993	PD	0.13 ↓	1.159	0.002
21	4-Methylphenol	C_7_H_8_O	108.1378	PD	0.05 ↓	1.178	0.01
22	Isorhamnetin-3-*O*-sophoroside	C_28_H_32_O_17_	640.5435	FD	3.29 ↑	1.148	0.005 × 10^−1^
23	Butin	C_15_H_12_O_5_	272.2528	FD	9.44 ↑	1.185	0.002 × 10^−1^
24	7-Methoxy-3-[1-(3-pyridyl) methylidene]-4-chromanone	C_16_H_13_NO_3_	267.2793	FD	2.41 ↑	1.142	0.005
25	Isorhamnetin-3-*O*-rutinoside-4′-*O*-glucoside	C_34_H_42_O_21_	786.6847	FD	2.50 ↑	1.147	0.006
26	3,4′-Dihydroxyflavone	C_15_H_10_O_4_	254.2375	FD	1.02 × 10^3^ ↑	1.185	0.002
27	6,7,8-Tetrahydroxy-5-methoxyflavone	C_16_H_12_O_6_	300.2629	FD	2.07 ↑	1.133	0.002
28	Naringenin	C_15_H_12_O_5_	272.2528	FD	9.79 ↑	1.184	0.001
29	Quercetin-3-O-(6″-*O*-p-coumaroyl) sophoroside-7-*O*-rhamnoside	C_42_H_46_O_23_	918.8008	FD	3.72 ↑	1.175	0.014
30	Quercetin-3-O-glucosyl (1→4) rhamnoside-7-*O*-rutinoside	C_39_H_50_O_25_	918.7993	FD	3.12 ↑	1.165	0.015
31	5,6,7,3′,4′,5′-hexanmethoxyflavone	C_21_H_22_O_8_	402.3946	FD	5.91 ↑	1.181	0.002
32	Eriodictyol-3′-*O*-glucoside	C_21_H_22_O_11_	450.3928	FD	3.88 ↑	1.169	0.004
33	1,8-dihydroxy-2,6-dimethylxanthen-9-one	C_15_H_12_O_4_	256.2534	FD	1.34 × 10^4^ ↑	1.185	0.011
34	Pinocembrin	C_15_H_12_O_4_	256.2534	FD	4.11 ↑	1.129	0.012
35	Quercetin-3,7-di-*O*-glucoside	C_27_H_30_O_17_	626.5169	FD	2.57 × 10^3^ ↑	1.184	0.015
36	Naringenin-4′-*O*-glucoside	C_21_H_22_O_10_	434.3934	FD	2.02 ↑	1.163	0.015
37	Trilobatin	C_21_H_24_O_10_	436.4093	FD	2.43 ↑	1.039	0.007
38	Isorhamnetin-3-*O*-sophoroside-7-*O*-rhamnoside	C_34_H_42_O_21_	786.6847	FD	4.43 ↑	1.147	0.009
FD4 vs. FE5							
39	Tangeretin	C_20_H_20_O_7_	372.3686	FD	3.43 ↑	1.215	0.002 × 10^−1^
40	Vanillic acid	C_8_H_8_O_4_	168.1467	PD	0.45 ↓	1.215	0.004 × 10^−1^
41	4-Hydroxybenzoic acid	C_7_H_6_O_3_	138.1207	PD	0.48 ↓	1.211	0.001
42	Methyl 3-(4-hydroxyphenyl) propionate	C_10_H_12_O_3_	180.2005	PD	6.99 × 10^−4^ ↓	1.208	0.010
43	3,5,7,3′4′-Pentamethoxyflavone	C_20_H_20_O_7_	372.3686	FD	2.87 ↑	1.205	0.003 × 10^−1^
44	Ferulic acid methyl ester	C_11_H_12_O_4_	208.2106	PD	0.25 ↓	1.199	0.002
45	2,5-Dihydroxybenzaldehyde	C_7_H_6_O_3_	138.1207	PD	7.29 × 10^−5^ ↓	1.193	0.001
46	Apigenin-7,4′-dimethyl ether	C_17_H_14_O_5_	298.2901	FD	3.11 ↑	1.189	0.004
47	Methyl syringate	C_10_H_12_O_5_	212.1993	PD	0.39 ↓	1.173	0.030
48	3-(3-Hydroxyphenyl)-3-hydroxypropanoic acid	C_9_H_10_O_4_	182.1733	PD	0.42 ↓	1.145	0.024

↑: fold change ≥ 2; ↓: fold change ≤ 0.5; PD: phenolic acid; FD: flavonoid.

## Data Availability

Data are contained within the article or Appendix A.

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
