# Peer review of "Plant-Wide Target Metabolomics Provides a Novel Interpretation of the Changes in Chemical Components during Dendrobium officinale Traditional Processing"

_antioxidants, 2023, doi:10.3390/antiox12111995_

Round 1
Reviewer 1 Report
Comments and Suggestions for Authors
This MS try to analyze the metabolomics to interpret the chemical changes of Dendrobium officinale during five steps of processing. According the previous paper by the similar authors, naringenin, 4,4’dihydroxy-3-5-dimethydibenzyl, norbilin A etc. were higher in the Fengdou and might be the potential hypoglycemic functional components.
1. However, in this MS, the purpose is not clear. Only “ small space and good keeping qualities” was mentioned (L47), but what is the quality? The kind of function that target metabolomics were connected with should be listed clearly.
2. Too many data, figures and tables were presented. It makes this MS lost focal point and quite blur, difficult to read.
3. All the figures are very bad in the resolution, very hard to understand.
4. PCA (L219-220) should point out the major contributor and discuss to match the result.
5. Fig. 3A-D is hard to match the data with the text. And Fig. 3E should be described in more detail.
6. Fig. S1 is not readable.
7. Table S1 (p8 to p35), with very small character, is not readable.
8. Changes in phenolic acids profile (L241-L306, Fig. S4 A-H) seems to be important, and should be put into the MS not in the supplementary figures.
9. It would be more clear if Table 1 can be concised and classified by the five step of processing.
10. The caption of Fig.6 should be deleted and changed into Fig.5.
Author Response
1. This MS try to analyze the metabolomics to interpret the chemical changes of Dendrobium officinale during five steps of processing. According the previous paper by the similar authors, naringenin, 4,4’dihydroxy-3-5-dimethydibenzyl, norbilin A etc. were higher in the Fengdou and might be the potential hypoglycemic functional components. However, in this MS, the purpose is not clear. Only “small space and good keeping qualities” was mentioned (L47), but what is the quality? The kind of function that target metabolomics were connected with should be listed clearly.
Response: Thanks for your suggestion. The chemical components of Dendrobium officinale mainly include polysaccharides, phenols, alkaloids, amino acids, etc. After processing the fresh strips of Dendrobium officinale into Fengdou, in addition to "small space and good keeping qualities", its chemical composition and antioxidant and hypoglycemic activities are significantly increased. We attempted to explore the hypoglycemic and antioxidant active ingredients by studying the activity and compositional differences between Dendrobium officinale fresh strips and maple dipper. Our previous study found that flavonoids and bibenzyl compounds such as naringenin, 4,4’dihydroxy-3-5-dimethydibenzyl, norbilin A, and dendrobenol were found to be higher in maple dipper, which may be the active components of Dendrobium officinale in lowering blood glucose [1]. In addition to its good hypoglycemic activity, Dendrobium officinale has a good antioxidant effect, and oxidative stress is a central pathogenetic mechanism leading to the chronic complications of diabetes mellitus [2]. However, we focused on mining the antioxidant active ingredients in this paper and found that the content of p-coumaric acid increased after processing of maple dipper, in which p-coumaric acid is also a hypoglycemic active ingredient.
[1] Liu, P.F.; Fan, B.; Liu, X.D.; Yang, Y.; Lu, C.; Tong, L.T.; Sun, J.; Wang, F.Z. Comparison of different components of Dendrobium officinale Fengdou before and after processing. J. Nucl. Agric. Biol. 2022, 36(12), 2412–2418.
[2] Zhang, P.; Li, T.; Wu, X.; Nice, E.C.; Huang, C.; Zhang, Y. Oxidative stress and diabetes: antioxidative strategies. Front Med. 2020, 14(5):583–600.
2. Too many data, figures and tables were presented. It makes this MS lost focal point and quite blur, difficult to read.
Response: Thanks for your suggestion. We have changed the previous figure 5 in the manuscript to Supplementary Figure S4. The data, picture presentation and order of placement in this paper have been adjusted to address the focus of the study in this MS.
3. All the figures are very bad in the resolution, very hard to understand.
Response: Sorry for our carelessness. The resolution ratios of all figures have been adjusted to 768psi, and they were uploaded as original files without compression.
4. PCA (L219-220) should point out the major contributor and discuss to match the result.
Response: Discussion regarding the PCA charts has been added to the manuscript, please see lines 214~221.
5. Fig. 3A-D is hard to match the data with the text. And Fig. 3E should be described in more detail.
Response: Thanks for your suggestion. This is our carelessness, and we have corrected it in the article. In the volcano plots, we compared the FA1~FE5 groupings two by two, with red indicating upward and green indicating downward adjustments. The most significant differences in metabolites (271 up and 210 down) were evident in the FA1 vs. FB2 group, suggesting that temperature is the key factor. In contrast, the least number of differential metabolites (only 67) was revealed in the FC3 vs. FD4 group, suggesting that stereotyping has little effect on composition. It also confirms the previous conclusion that temperature is a key factor affecting the changes of chemical ingredients. Please see L243~L258.
Fig. 3E is a Venn diagram between two groups, with the outermost number representing the specific differential chemical composition.
6. Fig. S1 is not readable.
Response: Fig. S1 is the OPLS DA diagram of comparison between two groups in five processes. We adjusted the font size and used Photoshop software to adjust the resolution of the figures to 768 psi.
7. Table S1 (p8 to p35), with very small character, is not readable.
Response: Thanks for your suggestion. We have adjusted the font of Table S1 in supplementary file.
8. Changes in phenolic acids profile (L241-L306, Fig. S4 A-H) seems to be important, and should be put into the MS not in the supplementary figures.
Response: Thanks for your suggestion. We have changed Figure. S4 A-H to Figure 6 and please refer to section 3.3.1 in the manuscript for details.
9. It would be more clear if Table 1 can be concised and classified by the five step of processing.
Response: Thanks for your suggestion. Table 1 has been succinctly categorized by the five processing steps, as detailed in the manuscript with changes highlighted in red.
10. The caption of Fig.6 should be deleted and changed into Fig.5.
Response: Thanks for your suggestion. We have corrected the serial numbers of all the figures and confirmed them.

Reviewer 2 Report
Comments and Suggestions for Authors
The manuscript titled "Plant-Wide Target Metabolomics Provides a Novel Interpretation of the Changes in Chemical Ingredients during Dendrobium officinale Traditional Processing," authored by Pengfei Liu and colleagues, delves into the investigation of potential alterations in the functional bioactive compounds within Dendrobium officinale following five distinct processing treatments.
The manuscript is intriguing, harboring data that has the potential to significantly advance scientific knowledge. However, some revisions are necessary to align it for publication in Antioxidants.
In summary:
The authors frequently employ the term "ingredients" to refer to bioactive components that might be found within the plant matrix. This term is highly technical and is typically used in the realm of food processing to denote components of a formulation or in pharmaceuticals to describe elements of a preparation. In the context of this manuscript, it does not appear to be the most suitable choice. The authors should consider replacing this term with a more context-appropriate alternative.
A change in the keywords is advisable. Keywords serve the purpose of enhancing the discoverability of the manuscript once it's published through widely used search engines like PubMed and Google Scholar. These tools rely on algorithms to identify articles based on words found in the title, abstract, or keywords. Consequently, this section should feature keywords that describe the work but are not already part of the title. Additionally, it's worth noting that the journal allows a maximum of 10 keywords. It is strongly recommended to make alterations and maximize the inclusion of keywords.
Within the introduction, the authors should provide a more comprehensive description of the plant's use in traditional medicine and underscore the significance of bioactive compounds within the plant matrix. Additionally, they should incorporate information on the plant's economic impact and, if applicable, its role in the nutraceutical industry.
Why is Figure 1 placed in the Materials and Methods section? This figure comprises four panels with data, yet the methodology hasn't been detailed yet. It would be prudent to segregate this figure by relocating panel A to the Materials and Methods section and situating all other panels in their respective sections within Results and Discussion.
Figure captions are currently lacking in detail. They should encompass more information regarding the content of the graphs, units of measurement, and the methodology employed for quantifying compounds, among other details. Furthermore, many figures lack a statistical analysis, which should be performed. Information pertaining to any statistical analyses conducted should be included in the figure captions.
At line 113, the authors present an equation within the text. As per the journal's guidelines, equations should be integrated into the text using Word's equation tool, with each equation assigned a number. Additionally, every equation should be cited in the text. It is strongly advised to consult the journal's author guidelines for a more comprehensive understanding of this aspect.
In the Materials and Methods section, the subsection on qualitative analyses should be distinct from that concerning statistical analyses.
The whereabouts of Figure 6 remain unclear.
A notable omission in the Results and Discussion section is the comparative analysis with other data reported in the existing literature. The authors have aptly cited and referenced articles to elucidate their results. However, given that there's no constraint on the number of references permissible in a scientific article, it is highly recommended to include additional references for the purpose of contrasting your results with those previously documented.
Comments on the Quality of English LanguageThe manuscript is, on the whole, clear and comprehensible. Nonetheless, numerous terms employed are ill-suited for a scientific discourse. Additionally, the structure of several sentences is flawed. A moderate English revision is warranted.
Author Response
1. The authors frequently employ the term "ingredients" to refer to bioactive components that might be found within the plant matrix. This term is highly technical and is typically used in the realm of food processing to denote components of a formulation or in pharmaceuticals to describe elements of a preparation. In the context of this manuscript, it does not appear to be the most suitable choice. The authors should consider replacing this term with a more context-appropriate alternative.
Response: Thanks for your suggestion. The term “ingredients” has been replaced by “chemical compositions” throughout the manuscript.
2. A change in the keywords is advisable. Keywords serve the purpose of enhancing the discoverability of the manuscript once it's published through widely used search engines like PubMed and Google Scholar. These tools rely on algorithms to identify articles based on words found in the title, abstract, or keywords. Consequently, this section should feature keywords that describe the work but are not already part of the title. Additionally, it's worth noting that the journal allows a maximum of 10 keywords. It is strongly recommended to make alterations and maximize the inclusion of keywords.
Response: Thanks for your suggestion. The key words for this manuscript have been revised as follows: Dendrobium officinale, traditional processing, widely-targeted metabolic analysis, chemical compositions, antioxidant activity.
3. Within the introduction, the authors should provide a more comprehensive description of the plant's use in traditional medicine and underscore the significance of bioactive compounds within the plant matrix. Additionally, they should incorporate information on the plant's economic impact and, if applicable, its role in the nutraceutical industry.
Response: Thanks for your suggestion. We have elaborated on the use of Dendrobium officinale in traditional medicine, its economic impact, and its role in the nutraceutical industry in the introduction section of the manuscript. For further information, please see L29~L48.
4. Why is Figure 1 placed in the Materials and Methods section? This figure comprises four panels with data, yet the methodology hasn't been detailed yet. It would be prudent to segregate this figure by relocating panel A to the Materials and Methods section and situating all other panels in their respective sections within Results and Discussion.
Response: Thanks for your suggestion. Figure 1 has been divided with panel A relocated to the Materials and Methods section, while all other panels have been situated in their respective sections within Results and Discussion.
5. Figure captions are currently lacking in detail. They should encompass more information regarding the content of the graphs, units of measurement, and the methodology employed for quantifying compounds, among other details. Furthermore, many figures lack a statistical analysis, which should be performed. Information pertaining to any statistical analyses conducted should be included in the figure captions.
Response: Thanks for your suggestion. We have revised some of the figure captions to include more information; the details can be seen in the red text in the manuscript.
6. At line 113, the authors present an equation within the text. As per the journal's guidelines, equations should be integrated into the text using Word's equation tool, with each equation assigned a number. Additionally, every equation should be cited in the text. It is strongly advised to consult the journal's author guidelines for a more comprehensive understanding of this aspect.
Response: Thanks for your suggestion. We have ensured that the equation within the text is formatted using Word's equation tool, with each equation assigned a number.
7. In the Materials and Methods section, the subsection on qualitative analyses should be distinct from that concerning statistical analyses.
Response: Thanks for your suggestion. The subsections on qualitative analyses and statistical analyses have been revised and separated into distinct sections in the manuscript. For more information, please see L150~L172.
8. The whereabouts of Figure 6 remain unclear.
Response: Thanks for your suggestion. The original Figure 6 has been modified and adjusted to become Figure 7, which is analyzed in detail in the manuscript on lines 414~415.
9. A notable omission in the Results and Discussion section is the comparative analysis with other data reported in the existing literature. The authors have aptly cited and referenced articles to elucidate their results. However, given that there's no constraint on the number of references permissible in a scientific article, it is highly recommended to include additional references for the purpose of contrasting your results with those previously documented.
Response: Thanks for your suggestion. We have cited additional literature in the Results and Discussion sections of the manuscript for the purpose of contrasting our results with those previously documented. Please see L548~L550 and L556~557.

Round 2
Reviewer 1 Report
Comments and Suggestions for Authors
The revised MS is much more clear now. But still with some small defect.
1. In L119, “as” should be “AS”
2. The typesetting of Table 1 may be rearranged in a better way.
Reviewer 2 Report
Comments and Suggestions for Authors
The authors revised the manuscript following the suggestions. now the manuscript is suitable for the publication.
Author Response
Thank you for your affirmation.